# Ionized calcium level at emergency department arrival is associated with return of spontaneous circulation in out-of-hospital cardiac arrest

**Sun Ju Kim**[1], **Hye Sim Kim**[2], **Sung Oh Hwang**[1], **Woo Jin Jung**[1], **Young Il Roh**[1], **Kyoung-Chul Cha**[1]\*, **Sang Do Shin**[3], **Kyoung Jun Song**[4], **on behalf of the Korean Cardiac Arrest Research Consortium (KoCARC) Investigators**[¶]

1 Department of Emergency Medicine, Yonsei University Wonju College of Medicine, Wonju, Republic of Korea, 2 Center of Biomedical Data Science, Yonsei University Wonju College of Medicine, Wonju, Republic of Korea, 3 Department of Emergency Medicine, Seoul National University Hospital, Seoul, Republic of Korea, 4 Department of Emergency Medicine, Seoul National University Boramae Medical Center, Seoul, Republic of Korea

¶ Membership of the Korean Cardiac Arrest Research Consortium (KoCARC) Investigators is provided in the Acknowledgments.
\* chaemp@yonsei.ac.kr

**Data Availability Statement:** Data cannot be shared publicly because of consent of personal information. Data are available from the Korean Cardiac Arrest Research Consortium data registry

## Abstract

### Background

Calcium level is associated with sudden cardiac death based on several cohort studies. However, there is limited evidence on the association between ionized calcium, active form of calcium, and resuscitation outcome. This study aimed to evaluate the potential role of ionized calcium in predicting resuscitation outcome in patients with out-of-hospital cardiac arrest.

### Methods

We analyzed the Korean Cardiac Arrest Research Consortium data (KoCARC) registry, a web-based multicenter registry that included 65 participating hospitals throughout the Republic of Korea. The patients with out-of-hospital cardiac arrest over 19 years old and acquired laboratory data including calcium, ionized calcium, potassium, phosphorus, creatinine, albumin at emergency department (ED) arrival were included. The primary outcome was successful rate of return of spontaneous circulation (ROSC) and the secondary outcomes were survival hospital discharge and favorable neurological outcome (cerebral performance category 1 or 2) at hospital discharge.

### Results

Eight-hundred and eighty-three patients were enrolled in the final analysis and 448 cases (54%) had ROSC. In multivariable logistic regression analysis, ionized calcium level was associated with ROSC (odds ratio, 1.77; 95% CI1.28–2.45; p = 0.001) even though calcium

(KoCARC) commitee. (ClinicalTrials.gov, number NCT03222999) The data can be accessed under the permission from Data Access Committee of KoCARC registry. The contact information is as follows; E-mail address: kocarc_cc@naver.com.

**Funding:** This study was supported by the Korea Centers for Disease Control and Prevention.

**Competing interests:** All authors declare that they have no competing interests.

level was not associated with ROSC (odds ratio, 0.87; 95% CI 0.70–1.08; $p = 0.199$). However, ionized calcium level was not associated with survival discharge (odds ratio, 0.99; 95% CI 0.72–1.36; $p = 0.948$) or favorable neurologic outcome (odds ratio, 0.45; 95% CI 0.03–6.55, $p = 0.560$).

## Conclusion

A high ionized calcium level measured during cardiopulmonary resuscitation was associated with an increased likelihood of ROSC.

## Introduction

Low serum calcium level is associated with the development of sudden cardiac death [1, 2]. Hypocalcemia can cause prolongation of QT interval, resulting in torsade de pointes and cardiac arrest, so it is should be properly managed through the administration of calcium chloride or calcium gluconate [3]. Calcium administration in the treatment of patients with cardiac arrest was initially recommended by the American Heart Association Guidelines for advanced life support in 1974 [4]. At that time, it was recommended in patients with any type of rhythm based on the physiologic effect of calcium on cardiac contractility, not on clinical evidence. Therefore, the recommendation was withdrawn since the establishment of cardiopulmonary resuscitation (CPR) guidelines in 2000 because there were studies against the use of calcium during resuscitation, and this has not been changed despite the references being small population-based studies with low level of evidence [5, 6]. However, the effect of calcium misinterpreted because the administration of calcium in these studies was not based on the serum calcium level during CPR. Furthermore, ionized calcium level is a better parameter than total calcium level in monitoring or treating a patient needing calcium replacement. Thus, monitoring the ionized calcium level might be helpful in maintaining optimal cardiac contractility in a patient with cardiac arrest.

In the Republic of Korea, research collaborators have conducted a large-population-based multicenter cohort study on out-of-hospital cardiac arrest, including total calcium and ionized calcium levels at emergency department (ED) arrival. These calcium levels could be associated with the cause or prognosis of cardiac arrest because these are collected immediately after the occurrence of cardiac arrest. We conducted a study to evaluate the potential role of ionized calcium level on resuscitation outcomes in a patient with out-of-hospital cardiac arrest using a Korean registry.

## Methods

### Data source

This was a registry-based, prospective observational study that analyzed the Korean Cardiac Arrest Research Consortium (KoCARC) registry data between 2014 and 2018. The KoCARC registry is a web-based multicenter registry including 64 participating hospitals throughout the Republic of Korea (ClinicalTrials.gov, number NCT03222999).

Variables in the KoCARC registry include patient information (e.g., age, sex, medical history, do-not-resuscitate information, and witness of cardiac arrest), community and prehospital resuscitation (e.g., place, time, etiology of cardiac arrest, existence of bystander, bystander CPR, emergency medical service resuscitation, prehospital defibrillation, and resuscitation

duration at scene and during transportation), hospital resuscitation (e.g., advanced airway, total administered dose of epinephrine, frequency of defibrillation, and laboratory tests at ED arrival), post-resuscitation care (e.g., targeted temperature management (TTM), vasopressor administration, and coronary intervention), and patient outcomes (e.g., return of spontaneous circulation (ROSC), survival to hospital discharge, and neurologic outcome at hospital discharge and 6 months after cardiac arrest occurrence) [7].

In all participating hospitals, the laboratory test was conducted upon ED arrival and optional at the time of KoCARC registry establishment (October 2015), but it was changed to obligatory variables since July 2017. The test variables were as follows: white blood cell count; hemoglobin count; platelet count; sodium, potassium, blood urea nitrogen (BUN), creatinine, aspartate aminotransferase, alanine aminotransferase, total bilirubin, albumin, calcium, ionized calcium, magnesium, phosphorous, total protein, glucose, total cholesterol, B-type natriuretic peptide, and d-dimer levels; and prothrombin time. Arterial blood gas analysis, including partial pressure of oxygen, partial pressure of carbon dioxide, base excess, arterial saturation, and lactate level, was also performed.

The Data Safety and Monitoring Board Committee of the KoCARC was organized to provide data quality control.

## Study variables

The following demographic, clinical, and laboratory parameters were obtained from the KoCARC registry: age; sex; total CPR duration; estimated time from collapse to ED arrival; witness of cardiac arrest; bystander CPR; initial presenting rhythm; total administered dose of epinephrine; and blood tests acquired at ED arrival, including calcium, ionized calcium, and variables parameters known to affect calcium level, such as creatinine, potassium, BUN, magnesium, phosphorus, and albumin levels and arterial pH [8]. Data on TTM, survival to discharge, and favorable neurologic outcome were also collected. Estimated time from collapse to ED arrival was obtained by evaluating the time gap from collapse to blood sampling, and favorable neurologic outcome was defined as having a cerebral performance category score of 1 or 2.

This study protocol was approved by the Institutional Review Board of Wonju Severance Christian Hospital (IRB No.CR319065).

## Study endpoints

The primary outcome was the ROSC rate, and secondary outcomes were survival to hospital discharge and favorable neurologic outcome at hospital discharge.

## Statistical analysis

To compare the characteristics between the ROSC and non-ROSC groups, two-sample t-test was used for continuous variables, and the chi-square test or Fisher's exact test was used to compare categorical variables. To analyze the factors associated with ROSC, survival to discharge, and favorable neurologic outcome, univariable and multivariable logistic regression analyses were performed, and cubic spline was fitted to estimate the odds ratio (OR).

Analyses were performed using the SAS program (version 9.4, SAS Institute Inc., Cary, NC, USA). A P-value < 0.05 was considered statistically significant.

## Results

### General characteristics

During the study period, 7,525 patients were enrolled in the KoCARC registry. Patients who were transferred from other hospitals (n = 1,251), aged <19 years (n = 177), with a do-not-resuscitate order (n = 477), with insufficient data (n = 119), and with missed laboratory data (n = 4,670) were excluded (S1 Fig). Finally, 831 patients were included in the final analysis.

There were 545 (66%) men, and the mean age was 68 (±15) years. The total CPR duration and estimated time from collapse to ED arrival were longer in the non-ROSC group (p = 0.001 and p<0.001, respectively). Witnessed cardiac arrest and bystander CPR were more frequently observed in the ROSC group (p<0.001). Regarding the initial presenting rhythm, ventricular fibrillation and pulseless ventricular tachycardia were more frequently observed in the ROSC group (p<0.001), but the total administered dose of epinephrine was higher in the non-ROSC group (p<0.001). In the laboratory tests, potassium (p = 0.020), calcium (p = 0.015), and magnesium (p = 0.015) levels were higher in the non-ROSC group, whereas ionized calcium level was higher in the ROSC group (p<0.001). TTM was performed in all patients with ROSC (Table 1).

### Factors associated with ROSC

In the univariable logistic regression analysis, factors associated with ROSC were verified, and the result is shown in Table 2. Total CPR duration, estimated time from collapse to ED arrival, witnessed cardiac arrest, and total administered dose of epinephrine were associated with

**Table 1. General characteristics.**

| Variable | Total (N = 831) | Non-ROSC (n = 383) | ROSC (n = 448) | *P* value |
|---|---|---|---|---|
| Male sex, n (%) | 545 (65.6) | 253 (66.0) | 292 (65.2) | 0.790 |
| Age, year, mean ± SD | 68.3 ± 14.9 | 70.0 ± 14.6 | 66.8 ± 15.0 | 0.002 |
| Total CPR duration | 53.8 ± 90.7 | 65.4 ± 107.4 | 43.9 ± 72.8 | 0.001 |
| Estimated time from collapse to ED arrival (min), mean ± SD | 41.6 ± 70.5 | 53.30 ± 93.8 | 31.6 ± 38.4 | <0.001 |
| Witness of cardiac arrest, n (%) | 551 (66.3) | 221 (57.7) | 330 (73.7) | <0.001 |
| Bystander CPR, n (%) | 434 (52.4) | 104 (28.3) | 151 (34.7) | <0.001 |
| Initial presenting rhythm, n (%) | | | | <0.001 |
| VF/pVT | 134 (16.1) | 53 (13.8) | 81 (18.9) | |
| Pulseless electrical activity | 228 (27.4) | 79 (20.6) | 149 (33.3) | |
| Asystole | 469 (56.4) | 251 (65.5) | 218 (48.7) | |
| Total administered dose of epinephrine (mg), mean ± SD | 6.67 ± 5.0 | 8.5 ± 4.74 | 5.07 ± 4.71 | <0.001 |
| Creatinine level (mg/dL), mean ± SD | 2.32 ± 5.8 | 2.2 ± 3.3 | 2.42 ± 7.3 | 0.578 |
| Potassium level (mmol/L), mean ± SD | 6.15 ± 5.0 | 6.6 ± 2.2 | 5.8 ± 6.5 | 0.020 |
| BUN level (mg/dL), mean ± SD | 30.76 ± 29.0 | 33.0 ± 35.9 | 28.9 ± 21.5 | 0.053 |
| Calcium level (mg/dL), mean ± SD | 8.61 ± 1.4 | 8.75 ± 1.6 | 8.5 ± 1.2 | 0.015 |
| Ionized calcium level (mmol/L), mean ± SD | 2.00 ± 1.5 | 1.79 ± 1.4 | 2.2 ± 1.6 | <0.001 |
| Magnesium level (mEq/L), mean ± SD | 2.45 ± 0.8 | 2.53 ± 0.8 | 2.4 ± 0.8 | 0.015 |
| Phosphorus level (mg/dL), mean ± SD | 8.66 ± 8.0 | 8.73 ± 2.9 | 8.6 ± 10.6 | 0.847 |
| Albumin level (g/dL), mean ± SD | 3.43 ± 10.8 | 3.89 ± 15.9 | 3.0 ± 0.8 | 0.306 |
| Arterial pH (pH), mean ± SD | 7.01 ± 2.1 | 7.09 ± 3.1 | 7.0 ± 0.2 | 0.396 |
| TTM after ROSC, n (%) | | | 448 (100) | |

BUN, blood urea nitrogen; CPR, cardiopulmonary resuscitation; ED, emergency department; pVT, pulseless ventricular tachycardia; ROSC, return of spontaneous circulation; SD, standard deviation; TTM, targeted temperature management; VF, ventricular fibrillation. Significance level set at a *P* < 0.05.

**Table 2. Factors associated with ROSC in the univariate logistic regression analysis.**

| Variable | Odds ratio | 95% CI | *P* value |
|---|---|---|---|
| Age | 0.99 | 0.98–1.00 | 0.002 |
| Sex (ref. female) | 0.96 | 0.72–1.28 | 0.791 |
| Total CPR duration (min) | 0.98 | 0.97–0.99 | <0.001 |
| Estimated time from collapse to ED arrival (min) | 0.99 | 0.98–0.99 | <0.001 |
| Witness of cardiac arrest | 2.05 | 1.53–2.75 | <0.001 |
| Bystander CPR | 1.35 | 1.00–1.82 | 0.054 |
| Initial shockable rhythm | 1.37 | 0.94–2.00 | 0.098 |
| Total administered dose of epinephrine (mg) | 0.84 | 0.81–0.87 | <0.001 |
| Creatinine level (mg/dL) | 1.01 | 0.98–1.04 | 0.607 |
| Potassium level (mmol/L) | 0.95 | 0.90–1.00 | 0.069 |
| BUN level (mg/dL) | 1.00 | 0.99–1.00 | 0.051 |
| Calcium level (mg/dL) | 0.88 | 0.80–0.98 | 0.015 |
| Ionized calcium level (mmol/L) | 1.18 | 1.08–1.29 | <0.001 |
| Magnesium level (mEq/L) | 0.78 | 0.64–0.96 | 0.016 |
| Phosphorus level (mg/dL) | 1.00 | 0.98–1.02 | 0.847 |
| Albumin level (g/dL) | 0.96 | 0.80–1.16 | 0.665 |
| Arterial pH | 0.96 | 0.85–1.08 | 0.465 |

BUN, blood urea nitrogen; CI, confidence interval; CPR, cardiopulmonary resuscitation; ED, emergency department.

ROSC, but bystander CPR was not associated with it. In the laboratory test upon ED arrival, calcium, ionized calcium, and magnesium levels were associated with ROSC.

## Analysis of the effect of calcium or ionized calcium level at ED arrival on ROSC

The multivariable logistic regression analysis was performed to verify the effect of calcium or ionized calcium level at ED arrival on ROSC. Model 1 was created based on variables with a P-value <0.1 in the univariable logistic regression analysis. Model 2 was created based on variables known to affect the serum calcium or ionized calcium levels, such as creatinine, potassium, BUN, magnesium, phosphorus, and albumin levels and arterial pH. Models 1 and 2 were adjusted simultaneously in model 3. In adjusted model 3, the ionized calcium level was associated with ROSC (OR: 1.89, 95% CI: 1.35–2.66; p<0.001) even though the total calcium level was not associated with ROSC (OR: 0.87, 95% CI: 0.70–1.08; p = 0.199) (Tables 3 and 4). Cubic spline was fitted to visualize differences in the OR of ROSC according to ionized calcium level, and the difference in OR by sex was also analyzed. The OR of ROSC increased proportionally to the ionized calcium level, and this tendency was shown in both sexes (Fig 1).

## Relationship between survival to discharge and favorable neurologic outcome and ionized calcium level

Ionized calcium level was not associated with survival to discharge (OR: 0.99, 95% CI: 0.72–1.36; p = 0.948) or favorable neurologic outcome (OR: 0.45, 95% CI: 0.03–6.55; p = 0.560) (S1 Table).

## Discussion

The ionized calcium level at ED arrival was associated with successful ROSC in this study.

**Table 3. Correlation between calcium level and ROSC in the multivariate logistic regression analysis.**

| Model | Odds ratio | 95% CI | P value |
|---|---|---|---|
| Crude | 0.88 | 0.80–0.98 | 0.014 |
| Model 1[†] | 0.90 | 0.79–1.02 | 0.110 |
| Model 2[‡] | 0.88 | 0.73–1.06 | 0.171 |
| Model 3[§] | 0.87 | 0.70–1.08 | 0.199 |

CI, confidence interval; CPR, cardiopulmonary resuscitation; ED, emergency department; ROSC, return of spontaneous circulation.

[†]Adjusted for age, sex, total CPR duration, estimated time from collapse to ED arrival, witness of cardiac arrest, bystander CPR, and total administered epinephrine dose.

[‡]Adjusted for magnesium, albumin, phosphorus, blood urea nitrogen, and creatinine levels and arterial pH.

[§]Adjusted for Model 1 + Model 2.

Hypocalcemia can induce fatal arrhythmia or cardiac arrest, because calcium is an essential cation in the generation of myocardial action potential resulting in contraction of cardiac muscles and maintenance of vascular tone [9–11]. Therefore, the maintenance of optimal calcium level is important to maintain normal cardiac function and systemic perfusion [12]. There was trial to promote cardiac contractility during CPR based on above biochemical background, but it was withdrawn from CPR guidelines because of lack of evidence for improving resuscitation outcomes [4–6]. However, this recommendation was based on a small population based studies analyzing the relation between resuscitation outcomes and total calcium level, not the ionized calcium [13, 14]. Unfortunately, total calcium level is influenced by various conditions, such as hypoalbuminemia, azotemia, metabolic acidosis, hyperphosphatemia, lactic acidosis, and bicarbonate infusion [15, 16]. On the contrary, the free calcium cation, generally called ionized form, effect the movement between intracellular compartments and specific membrane protein pumps directly, which acts more important than other forms of calcium in the human body metabolism associated with calcium in physiology and biochemistry [8]. Therefore, it is recommended that the level of ionized calcium, a more reliable parameter and metabolized directly in humans, be monitored in clinical practice [17, 18]. We found the ionized calcium level at ED arrival was associated with ROSC and its probability was proportional to the ionized calcium level in this large-population-based observational study. It might imply that a prompt determination of the ionized calcium level at ED arrival and immediate infusion of calcium chloride or gluconate during CPR could promote ROSC [19, 20]. Considering

**Table 4. Relationship between ionized calcium level and ROSC in the multivariate logistic regression analysis.**

| Model | Odds ratio | 95% CI | P value |
|---|---|---|---|
| Crude | 1.18 | 1.08–1.29 | <0.001 |
| Model 1[†] | 1.19 | 1.06–1.34 | 0.003 |
| Model 2[‡] | 1.98 | 1.45–2.69 | <0.001 |
| Model 3[§] | 1.89 | 1.35–2.66 | <0.001 |

CI, confidence interval; CPR, cardiopulmonary resuscitation; ED, emergency department; ROSC, return of spontaneous circulation.

[†]Adjusted for age, sex, total CPR duration, estimated time from collapse to ED arrival, witness of cardiac arrest, bystander CPR, total administered epinephrine dose, and calcium level.

[‡]Adjusted for magnesium, albumin, phosphorus, blood urea nitrogen, creatinine, and calcium levels and arterial pH.

[§]Adjusted for Model 1 + Model 2.

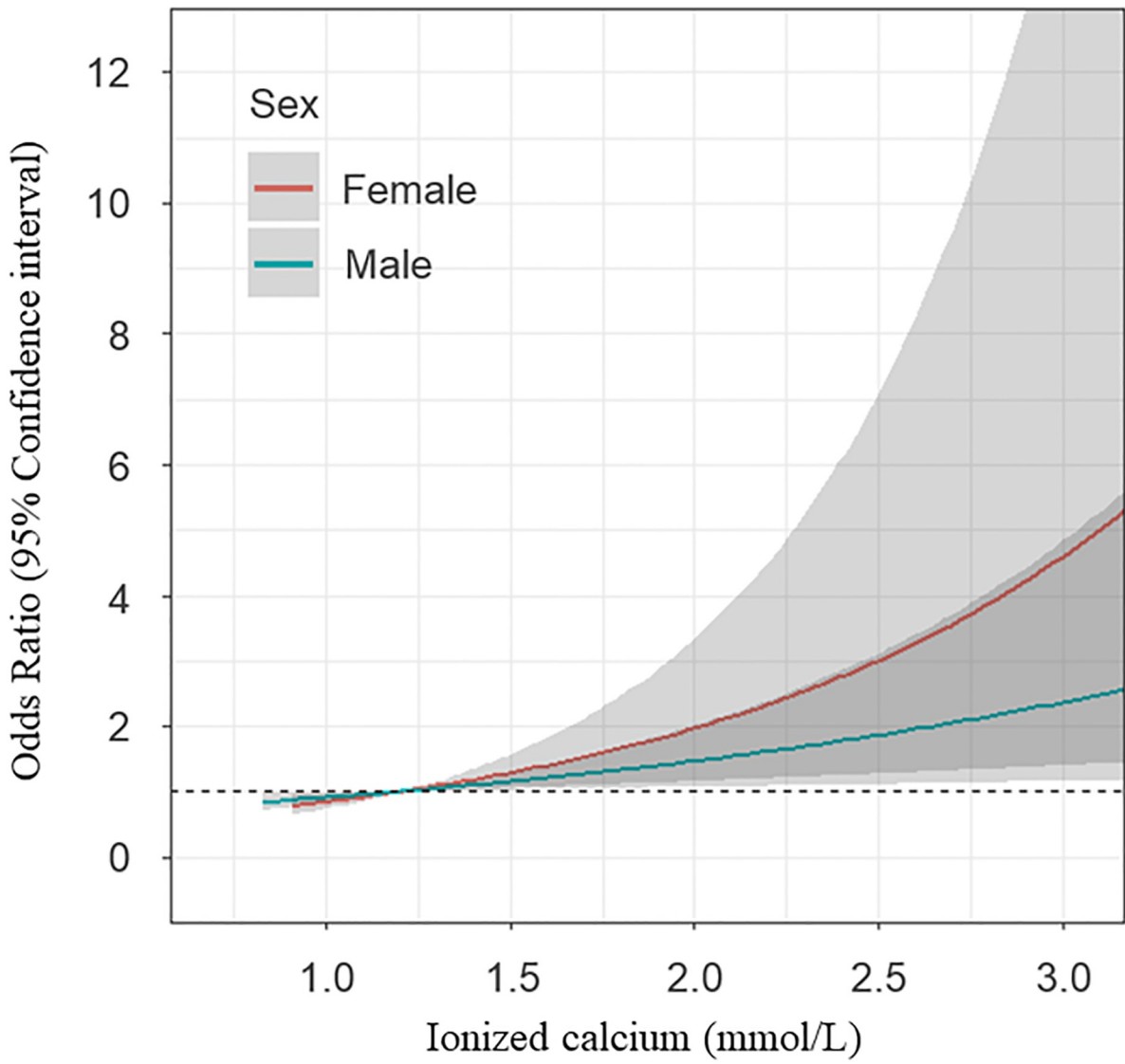

**Fig 1. The trend of odds ratio of return of spontaneous circulation followed by the ionized calcium.**

calcium infusion during CPR can be applicable because the ionized calcium level can be obtained in a short time, even during CPR using a point-of-care arterial blood analyzer widely used in ED or intensive care unit. Furthermore the effect of calcium can be observed immediately after infusion because calcium chloride or calcium gluconate can be infused intravenously and acts like ionized calcium without metabolism [21].

The ionized calcium level was not associated with survival to discharge and favorable neurologic outcome in this study. Because post-cardiac arrest care should be performed in patients with ROSC, most patients resuscitated successfully would be monitored and managed in the intensive care unit [22]. Electrolyte imbalance would be properly monitored and managed because it can promote poor prognosis [23, 24]. Therefore, it might be difficult to confirm survival to discharge with a single parameter such as ionized calcium level at ED arrival, which is why ionized calcium level was not associated with survival to discharge in this study. TTM is the most important treatment modality in promoting neurologic outcome and was performed

in all patients with ROSC in this study [25]. It would not affect neurologic outcomes in enrolled patients and was also the reason that ionized calcium level was not associated with favorable neurologic outcome in this study.

The administered epinephrine during CPR could change the level of ionized calcium. In a previous study, it was noticed that catecholamine could lower the calcium concentration [26]. However the opposite or neutral results from other animal studies were also reported and all above studies were not performed in patients with cardiac arrest [27, 28]. Therefore we couldn't figure out the relation between administered dose of epinephrine and the level of ionized calcium during resuscitation yet. We hope further study could verify the dose responsiveness of epinephrine for the level of ionized calcium in patient with cardiac arrest.

This study had several limitations. First, although this study was based on a relatively large population, selection bias might be present because laboratory tests were not performed in all patients registered in the KoCARC registry. Second, we did not account diseases that affect calcium homeostasis, such as parathyroid disease, in the medical history. Lastly, although all participating hospitals performed advanced life support following current CPR guidelines, additional calcium or sodium bicarbonate might be administered during resuscitation and could affect ROSC.

## Conclusion

The ionized calcium level at ED arrival is associated with ROSC. Future randomized controlled studies are needed to verify the precise effect of calcium infusion based on the ionized calcium level at ED arrival in promoting ROSC.

## Supporting information

**S1 Table. Correlation between ionized calcium concentration and survival discharge and favourable neurologic outcome by multivariable logistic regression test.**
(DOCX)

**S2 Table. The correlation analysis between total administered dose of epinephrine and the ionized calcium.**
(DOCX)

**S1 Fig. Patient flow of out-of-hospital cardiac arrest from 2014 to 2018 in KoCARC registry.** *KoCARC: Korean Cardiac Arrest Research Consortium data.
(TIF)

**S2 Fig. A scatter plot analysis between total administered dose of epinephrine and ionized calcium.**
(TIF)

## Acknowledgments

The Korean Cardiac Arrest Research Consortium was supported administratively by the Korea Centers for Disease Control and Prevention during the organizing stage.

We would like to acknowledge the chairman of the KoCARC: Sung Oh Hwang (Yonsei University Wonju College of Medicine, e-mail: sheang@yonsei.ac.kr) and members of the Secretariat: Jeong Ho Park (Seoul National University Hospital), Sun Young Lee (Seoul National University Hospital), Jung Eun Kim (Seoul National University Hospital), Na Young Kim (Seoul National University Hospital), and Min Ji Kwon (Seoul National University Hospital). We also thank the investigators from all participating hospitals in KoCARC: Myoung Chun

Kim (Kyung Hee University Hospital at Gangdong), Sang Kuk Han (Kangbuk Samsung Medical Center), Kwang Je Baek (Konkuk University Medical Center), Han Sung Choi (Kyung Hee University Hospital), Sung Hyuk Choi (Korea University Guro Hospital), Ik Joon Jo (Samsung Medical Center), Jong Whan Shin (SMG-SNU Boramae Medical Center), Sang Hyun Park (Seoul Medical Center), In Cheol Park (Yonsei University Severance Hospital), Chul Han (Ewha Womans University Mokdong Hospital), Chu Hyun Kim (Inje University Seoul Paik Hospital), Gu Hyun Kang (Hallym University Kangnam Sacred Heart Hospital), Tai Ho Im (Hanyang University Seoul Hospital), Seok Ran Yeom (Pusan National University Hospital), Jae Hoon Lee (Dong-a University Hospital), Ha Young Park (Inje University Haeundae Hospital), Jeong Bae Park (Kyungpook National University Hospital), Sung Jin Kim (Keimyung University Dongsan Medical Center), Kyung Woo Lee (Daegu Catholic University Medical Center), Woon Jeong Lee (The Catholic University of Korea Incheon ST. Mary's Hospital), Sung Hyun Yun (Catholic Kwandong University), Ah Jin Kim (Inha University Hospital), Kyung Woon Jeong (Chonnam National University Hospital), Sun Pyo Kim (Chosun University Hospital), Jin Woong Lee (Chungnam National University Hospital), Sung Soo Park (Konyang University Hospital), Ryeok Ahn (Konyang University Hospital), Kyoung Ho Choi (The Catholic University of Korea Uijeongbu St. Mary's Hospital), Young Gi Min (Ajou University Hospital), In Byung Kim (Myongji Hospital), Ji Hoon Kim (The Catholic University of Korea Buchen St. Mary's Hospital), Seung Chul Lee (Dongguk University Ilsan Hospital), Young Sik Kim (Bundang Jesaeng General Hospital), Hun Lim (Soonchunhyang University Bucheon Hospital), Jin Sik Park (Sejong Hospital), Jun Seok Park (Inje University Ilsan Paik Hospital), Dai Han Wi (Wonkwang University Sanbon Hospital), Ok Jun Kim (Cha University Bundang Cha Hospital), Bo Seung Kang (Hanyang University Guri Hospital), Soon Joo Wang (Hallym University Dongtan Sacred Heart Hospital), Se Hyun Oh (GangNeung Asan Hospital), Jun Hwi Cho (Kangwon National University Hospital), Mu Eob An (Hallym University Chuncheon Sacred Heart Hospital), Ji Han Lee (Chungbuk National University Hospital), Han Joo Choi (Dankook University Hospital), Jung Won Lee (Soonchunhyang University Cheonan Hospital), Tae Oh Jung (Chonbuk National University Hospital), Dai Hai Choi (Dongguk University Gyeongju Hospital), Seong Chun Kim (Gyeongsang National University Hospital), Ji Ho Ryu (Pusan National University Yangsan Hospital), Won Kim (Cheju Halla General Hospital), and Sung Wook Song (Jeju National University Hospital).

## Author Contributions

**Conceptualization:** Kyoung-Chul Cha.

**Data curation:** Sun Ju Kim, Woo Jin Jung, Young Il Roh, Kyoung-Chul Cha.

**Formal analysis:** Hye Sim Kim, Kyoung-Chul Cha.

**Methodology:** Sun Ju Kim, Kyoung-Chul Cha.

**Project administration:** Sang Do Shin.

**Resources:** Sang Do Shin, Kyoung Jun Song.

**Supervision:** Kyoung-Chul Cha.

**Writing – original draft:** Sun Ju Kim.

**Writing – review & editing:** Sung Oh Hwang, Woo Jin Jung, Kyoung-Chul Cha.

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
