## [Decision Letter · Decision Letter 0]

7 Aug 2020

PONE-D-20-12193

Ionized calcium level at emergency department arrival is associated with return of spontaneous circulation in out-of-hospital cardiac arrest

PLOS ONE

Dear Dr. Cha,

Thank you for submitting your manuscript to PLOS ONE. After careful consideration, we feel that it has merit but does not fully meet PLOS ONE’s publication criteria as it currently stands. Therefore, we invite you to submit a revised version of the manuscript that addresses the points raised during the review process.

We look forward to receiving your revised manuscript.

Kind regards,

Andrea Ballotta

Academic Editor

PLOS ONE

Additional Editor Comments:

Tx for having submitted your manuscript entitled "Ionized calcium level at emergency department arrival is associated with return of spontaneous circulation in out-of-hospital cardiac arrest". After careful consideration the two reviewers supported the option of acceptance for publication but just after addressing some minor issues.

Journal Requirements:

2. In ethics statement in the manuscript and in the online submission form, please provide additional information about the patient records used in your retrospective study. Specifically, please ensure that you have discussed whether all data were fully anonymized before you accessed them and/or whether the IRB or ethics committee waived the requirement for informed consent. If patients provided informed written consent to have data from their medical records used in research, please include this information.

"This consortium was supported by the Korea Centers for Disease Control and Prevention

during the organizing stage. Currently, the KoCARC is partly supported by the Korean

Association of Cardiopulmonary Resuscitation."

"The authors received no specific funding for this work."

5. One of the noted authors is a group or consortium [Korean Cardiac Arrest Research Consortium (KoCARC) Investigators]. In addition to naming the author group and listing the individual authors and affiliations within this group in the acknowledgments section of your manuscript, please also indicate clearly a lead author for this group along with a contact email address.

6. Please include a caption for figure 1.

Reviewers' comments:

Reviewer's Responses to Questions

**Comments to the Author**

1. Is the manuscript technically sound, and do the data support the conclusions?

Reviewer #1: Yes

Reviewer #2: Yes

2. Has the statistical analysis been performed appropriately and rigorously? 

Reviewer #1: Yes

Reviewer #2: Yes

3. Have the authors made all data underlying the findings in their manuscript fully available?

Reviewer #1: Yes

Reviewer #2: Yes

4. Is the manuscript presented in an intelligible fashion and written in standard English?

Reviewer #1: Yes

Reviewer #2: Yes

5. Review Comments to the Author

Reviewer #1: It's mine opinion that it's an interesting study that deserves further investigation useful to define the causes of non difference in terms of mortality.

Evalute treatment with calcium administration during cardiac arrest.

Reviewer #2: The authors have presented and addressed a potential important point in the field of cardiopulmonary resucitation where many uncertainty are still to be determined.

The paper is well written and message is clear with some practical insights.

Minor comments

Introduction line 63

might be misunderstood

Please change to “misled” or "misinterpreted" or "should be contextualised"

Line 70

Change “would” with “could”

“Relationship between survival to discharge and favorable  neurologic outcome and ionized calcium level”

It is hard to think that the first measurement of ionized calcium level in patients with ROSC may affect the outcome of the post-cardiac arrest syndrome. I would add this concept in the discussion.

Is it known How exogenous adrenaline administration affect ionized calcium? If yes, and the data are reliable, this should be referenced otherwise it could be a point for further analysis/study.

6. PLOS authors have the option to publish the peer review history of their article (what does this mean?). If published, this will include your full peer review and any attached files.

Reviewer #1: No

Reviewer #2: No

---

## [Author Response · Author response to Decision Letter 0]

27 Aug 2020

Response to reviewers

We appreciate your kind recommendations for improving the quality of our manuscript. Here we present our responses or revised content corresponding to your suggestions.

Comment 1. Please ensure that your manuscript meets PLOS ONE's style requirements, including those for file naming.

Answer 1: We checked it once more and confirmed the style requirement.

Comment 2. In ethics statement in the manuscript and in the online submission form, please provide additional information about the patient records used in your retrospective study. Specifically, please ensure that you have discussed whether all data were fully anonymized before you accessed them and/or whether the IRB or ethics committee waived the requirement for informed consent. If patients provided informed written consent to have data from their medical records used in research, please include this information.

Answer 2: We added the comments about data coding and informed consent as follows;

Patient information was coded as anonymous so that researchers could not recognize the patient’s personal information.

This study protocol was approved by the Institutional Review Board of Wonju Severance Christian Hospital (IRB No.CR319065) and informed consent was waived in case of unsuccessful resuscitation and obtained after intensive care unit admission in case of successful resuscitation.

 

Comment 3. Thank you for stating the following in the Acknowledgments Section of your manuscript:

"This consortium was supported by the Korea Centers for Disease Control and Prevention

during the organizing stage. Currently, the KoCARC is partly supported by the Korean

Association of Cardiopulmonary Resuscitation."

"The authors received no specific funding for this work."

Answer 3: We removed the funding statement from Acknowledgement section and added it in online submission form. We revised the comments about Korea Centers for Disease Control and Prevention because the institution supported administrative work, not a fund. The revised comment is as follows;

The Korean Cardiac Arrest Research Consortium was supported administratively by the Korea Centers for Disease Control and Prevention during the organizing stage.

Comment 4. We note that you have indicated that data from this study are available upon request. PLOS only allows data to be available upon request if there are legal or ethical restrictions on sharing data publicly. For information on unacceptable data access restrictions, please see http://journals.plos.org/plosone/s/data-availability#loc-unacceptable-data-access-restrictions.

Answer 4: We added the following policy for data access in the cover letter.

Comment 5. One of the noted authors is a group or consortium [Korean Cardiac Arrest Research Consortium (KoCARC) Investigators]. In addition to naming the author group and listing the individual authors and affiliations within this group in the acknowledgments section of your manuscript, please also indicate clearly a lead author for this group along with a contact email address.

Answer 5: We have indicated a chairman for our KoCARC registry with a contact e-mail address as follows;

We would like to acknowledge the chairman of the KoCARC: Sung Oh Hwang (Yonsei University Wonju College of Medicine, e-mail address: shwang@yonsei.ac.kr)

Comment 6. Please include a caption for figure 1.

Answer 6. We added a caption for fig 1.

Comment 7. Review Comments to the Author

Reviewer #1: It's mine opinion that it's an interesting study that deserves further investigation useful to define the causes of no difference in terms of mortality.

Evaluate treatment with calcium administration during cardiac arrest.

Reviewer #2: The authors have presented and addressed a potential important point in the field of cardiopulmonary resuscitation where many uncertainty are still to be determined.

The paper is well written and message is clear with some practical insights.

Minor comments

7-1: Introduction line 63

might be misunderstood

Please change to “misled” or "misinterpreted" or "should be contextualised"

Answer 7-1: Thank you for your comment. We corrected word as you recommended.

Comment 7-2: Line 70

Change “would” with “could”

Answer 7-2: Thank you for your comment. We corrected word as you recommended.

Comment 7-3: “Relationship between survival to discharge and favorable neurologic outcome and ionized calcium level”

It is hard to think that the first measurement of ionized calcium level in patients with ROSC may affect the outcome of the post-cardiac arrest syndrome. I would add this concept in the discussion.

Answer 7-3) Thank you very much for commenting on what we were worried about. As you mentioned, it is difficult to evaluate the predictability of the ionized calcium drawn at ED arrival for survival discharge or favorable neurologic outcome because there was high risk of bias from patient’s physiologic or pathologic status and treatment modalities or responsibility during post-cardiac arrest care. We removed the description about the relationship between ionized calcium and survival discharge or favorable neurologic outcome at first paragraph on discussion section and the related description was left on discussion section, line 233 through 243. 

Comment 7-4: Is it known How exogenous adrenaline administration affect ionized calcium? If yes, and the data are reliable, this should be referenced otherwise it could be a point for further analysis/study.

Answer 7-4: Thank you for your important comments. For confirming your suggestion, we drew a scatter plot and performed a correlation analysis between total administered dose of epinephrine and ionized calcium. 

In a scatter plot, there is no linear correlation between the two variables (Supplementary _ fig 2).

As a result of the correlation analysis, it was analyzed that there was a negative correlation between the two variables, but the correlation coefficient was close to 0, so there was little correlation between the two variables (R=-0.0135) (Supplementary_table 2).

There were some studies [1-3] on the relationship between administration of epinephrine and ionized calcium in non-cardiac arrest situation, but we cannot find a study in cardiac arrest situation.

Judging from descriptions above, it seems hard to believe that there is any correlation between total administered dose of epinephrine and the ionized calcium.

We added the above results on the supplements.

 

Reference

1. Kenny, A.D., Effect of catecholamines on serum calcium and phosphorus levels in intact and parathyroidectomized rats. Naunyn-Schmiedebergs Archiv für experimentelle Pathologie und Pharmakologie, 1964. 248(2): p. 144-152.

2. Ljunhgall, S., et al., Effects of epinephrine and norepinephrine on serum parathyroid hormone and calcium in normal subjects. Exp Clin Endocrinol, 1984. 84(3): p. 313-8.

3. Musso, E. and M. Vassalle, Effects of norepinephine, calcium, and rate of discharge on 42K movements in canine cardiac Purkinje fibers. Circulation research, 1978. 42(2): p. 276-284.

---

## [Decision Letter · Decision Letter 1]

28 Sep 2020

Ionized calcium level at emergency department arrival is associated with return of spontaneous circulation in out-of-hospital cardiac arrest

PONE-D-20-12193R1

Dear Dr. Cha,

We’re pleased to inform you that your manuscript has been judged scientifically suitable for publication and will be formally accepted for publication once it meets all outstanding technical requirements.

Kind regards,

Andrea Ballotta

Academic Editor

PLOS ONE

Reviewers' comments:

Reviewer's Responses to Questions

**Comments to the Author**

1. If the authors have adequately addressed your comments raised in a previous round of review and you feel that this manuscript is now acceptable for publication, you may indicate that here to bypass the “Comments to the Author” section, enter your conflict of interest statement in the “Confidential to Editor” section, and submit your "Accept" recommendation.

Reviewer #2: All comments have been addressed

2. Is the manuscript technically sound, and do the data support the conclusions?

Reviewer #2: Yes

3. Has the statistical analysis been performed appropriately and rigorously? 

Reviewer #2: Yes

4. Have the authors made all data underlying the findings in their manuscript fully available?

Reviewer #2: Yes

5. Is the manuscript presented in an intelligible fashion and written in standard English?

Reviewer #2: Yes

6. Review Comments to the Author

Reviewer #2: (No Response)

7. PLOS authors have the option to publish the peer review history of their article (what does this mean?). If published, this will include your full peer review and any attached files.

Reviewer #2: No

---

## [Editor Report · Acceptance letter]

2 Oct 2020

PONE-D-20-12193R1 

Ionized calcium level at emergency department arrival is associated with return of spontaneous circulation in out-of-hospital cardiac arrest 

Dear Dr. Cha:

I'm pleased to inform you that your manuscript has been deemed suitable for publication in PLOS ONE. Congratulations! Your manuscript is now with our production department. 

Kind regards, 

on behalf of

Dr. Andrea Ballotta 

Academic Editor

PLOS ONE